# Parity-independent Kondo effect of correlated electrons in electrostatically defined ZnO quantum dots

Kosuke Noro [1,2], Yusuke Kozuka [3], Kazuma Matsumura[1,2], Takeshi Kumasaka[1], Yoshihiro Fujiwara[1,2], Atsushi Tsukazaki [4,5], Masashi Kawasaki [6,7] & Tomohiro Otsuka [1,2,5,7,8] ✉

Quantum devices such as spin qubits have been extensively investigated in electrostatically confined quantum dots using high-quality semiconductor heterostructures like GaAs and Si. Here, we present a demonstration of electrostatically forming the quantum dots in ZnO heterostructures. Through the transport measurement, we uncover the distinctive signature of the Kondo effect independent of the even-odd electron number parity, which contrasts with the typical behavior of the Kondo effect in GaAs. By analyzing temperature and magnetic field dependences, we find that the absence of the even-odd parity in the Kondo effect is not straightforwardly interpreted by the considerations developed for conventional semiconductors. We propose that, based on the unique parameters of ZnO, electron correlation likely plays a fundamental role in this observation. Our study not only clarifies the physics of correlated electrons in the quantum dot but also holds promise for applications in quantum devices, leveraging the unique features of ZnO.

Advances in nanofabrication technology have allowed us to artificially create tiny semiconductor devices. A notable example is the semiconductor quantum dot, which confines electrons to a nanometer-scale area, enabling the direct control and observation of the quantized electronic states[1–3]. By precisely adjusting voltages on split gates, fundamental electronic properties of semiconductor quantum dots have been extensively investigated, encompassing orbital[1,4,5] and spin states[6–10]. Beyond single-particle properties, quantum dots serve as an ideal platform for exploring the physics of the quantum many-body effect, involving localized electrons and itinerant electrons surrounding them, which has unveiled interesting phenomena such as the Fano effect[11,12] and the Kondo effect[13–22]. Furthermore, because of the high controllability of the quantum states, quantum dots offer exciting prospects for quantum information devices, where electron spins are

used as qubits[23,24], as highly coherent manipulation[25–31] and their integration schemes[32–36] have been recently demonstrated.

Until now, semiconductor quantum dots have been actively studied in heterostructures employing materials like GaAs and Si. However, high-quality heterostructures fabricated from emergent semiconductors, such as graphene[37] and ZnO[38], have become available following prolonged efforts to develop manufacturing technologies. In ZnO heterostructures, which are the focus of this study, several intriguing phenomena resulting from the strong electron correlation have been reported, including quantum Hall ferromagnetic state[39], Winger crystallization[40,41], and even-denominator fractional quantum Hall effects[42,43]. Figure 1a summarizes the feature of ZnO compared to other semiconductor materials in terms of the electron interaction parameter $r_S$ and the transport scattering time $\tau$, where $r_S$ is defined as the

[1]Research Institute of Electrical Communication, Tohoku University, Sendai, Japan. [2]Department of Electronic Engineering, Graduate School of Engineering, Tohoku University, Sendai, Japan. [3]Research Center for Materials Nanoarchitectonics (MANA), National Institute for Material Science (NIMS), Tsukuba, Japan. [4]Institute for Materials Research, Tohoku University, Sendai, Japan. [5]Center for Science and Innovation in Spintronics, Tohoku University, Sendai, Japan. [6]Department of Applied Physics and Quantum-Phase Electronics Center (QPEC), University of Tokyo, Bunkyo-ku, Tokyo, Japan. [7]Center for Emergent Matter Science, RIKEN, Wako, Saitama, Japan. [8]WPI Advanced Institute for Materials Research, Tohoku University, Sendai, Japan. ✉e-mail: tomohiro.otsuka@tohoku.ac.jp

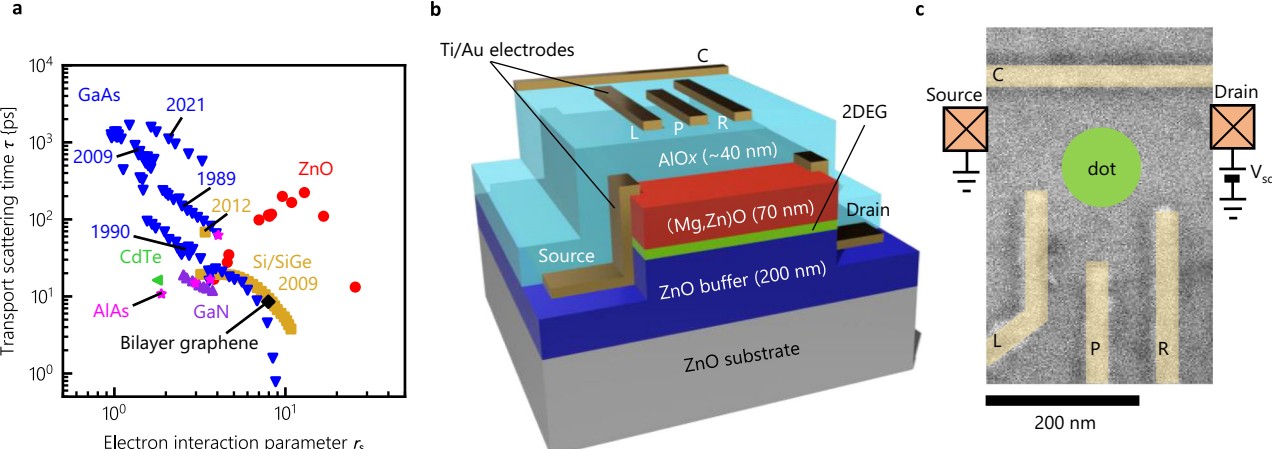

**Fig. 1 | ZnO and device structure. a** The map of material parameters in terms of the electron interaction parameter $r_S$ and the transport scattering time $\tau$, comparing ZnO and other semiconductors. **b** Schematic of the ZnO quantum dot device. Two-dimensional electron gas (2DEG) is formed at the (Mg,Zn)O/ZnO interface. Gate electrodes are fabricated on top of the AlO$_x$ gate insulator. **c** The false-colored SEM image of the ZnO quantum dot device.

ratio of the Coulomb energy to the Kinetic energy, and is expressed as $r_S = m^* e^2 / 4\pi\hbar^2 \varepsilon \sqrt{n\pi}$ ($m^*$: effective mass, $e$: elementary electric charge, $\hbar$:Planck constant divided by $2\pi$, $\varepsilon$:dielectric constant, $n$: sheet carrier density). ZnO combines strong electron correlation and clean transport, opening a new field of quantum dot research in strongly correlated systems. In addition to the correlation effect, ZnO stands out as a unique material compared to conventional semiconductors, characterized by its large band gap ($E_g$ = 3.37 eV)[44] with a single electron pocket preventing intervalley carrier scattering, weak spin-orbit interaction, and low-density nuclear spins ($^{67}$Zn (4% natural abundance) has a $I = 5/2$ nuclear spin, and $^{17}$O (0.04% natural abundance) has a $I = 5/2$ nuclear spin, while other Zn and O isotopes show zero nuclear spin states.).These features make ZnO suitable for quantum applications that leverage long spin coherence. Although spin-orbit interaction can be used for spin manipulation, it also causes decoherence and the appropriate control of the interaction is crucial. In Si spin qubits, introducing controllable effective spin interaction induced by micro-magnets in small spin-orbit materials are widely used[28]. Because of the small spin-orbit interaction in ZnO, the same approach can be employed.

Here, we demonstrate the electrostatic formation of quantum dots in high-quality ZnO heterostructures. By precisely tuning the split gate voltages, we observe Coulomb peaks and Coulomb diamonds at dilution temperatures, illustrating well-defined quantized states. Moreover, we identify the Kondo effect, characterized by zero-bias resonance peaks in the Coulomb diamonds. Remarkably, the Kondo effect proves to be resilient, independent of the even-odd electron number parity. This is in contrast to the commonly observed odd-parity Kondo effect in GaAs quantum dots when an unpaired localized spin is present. Our study shows the demonstration of electrostatically formed quantum dots in ZnO, shedding light on the fundamental properties of correlated localized electrons in the quantum dots.

## Results
### Device structure
The (Mg,Zn)O/ZnO heterostructure is grown on a Zn-polar ZnO (0001) substrate by molecular beam epitaxy, the details of which are explained in ref. 45 and "Methods". As shown in Fig. 1b, the two-dimensional electron gas (2DEG) forms at the interface between (Mg,Zn)O and ZnO. The density is modulated by applying the gate voltages across the AlO$_x$ insulator as described in ref. 46,47. The values of electron density and mobility without gate voltages are determined by the Hall measurement at 1.8 K as $n = 4.9 \times 10^{11}$ cm$^{-2}$ and

$\mu = 170,000$ cm$^2$ V$^{-1}$ s$^{-1}$, respectively. Figure 1c shows a false-colored scanning electron microscope (SEM) image of the planar structure of the top gates[48]. Here, the mean free path is estimated as ~2 μm, which is much larger than the gate structure. A quantum dot is formed by applying negative gate voltages on the gate electrodes C, L, P, and R, denoted as $V_C$, $V_L$, $V_P$, and $V_R$, respectively.

### Quantum dot formation and control
We first measure the electron transport through the device at a cryogenic temperature of 60 mK. Throughout the measurement, we set $V_C = -4.5$ V to fully deplete the electrons under gate C. Figure 2a shows a conductance map (in the unit of $2e^2/h$, $e$: elementary electric charge, $h$: Planck constant) while sweeping $V_R$ and $V_L$ that are applied to gate R and gate L with a fixed plunger gate voltage ($V_P$) of −5.0 V and a source-drain bias ($V_{sd}$) of 0.24 mV, where the electrons under the plunger gate (P) is also depleted. For $V_L > -4.0$ V and $V_R > -4.4$ V (top right in Fig. 2a), relatively high conductance is obtained, meaning that the transmission of electrons is large through the gaps between gates C and L and gates C and R. As we lower $V_R$ (Fig. 2b) and $V_L$ (Fig. 2c) along the orange line at top and the green line at right in Fig. 2a, the conductance decreases accompanied with pronounced oscillation patterns. These oscillations correspond to Coulomb oscillations associated with forming the quantized states separated by the on-site Coulomb energy plus the single-particle orbital energy in the quantum dot. Then, at fixed $V_L = -4.37$ V and $V_R = -4.79$ V (red cross in Fig. 2a), we control the number of electrons in the dot by sweeping $V_P$ as evidenced by the Coulomb oscillations in Fig. 2d. The width of the Coulomb peaks becomes wider at higher $V_P$, reflecting the increase of the tunnel coupling between the dot and the reservoir. The dot states are further confirmed by measuring the conductance as functions of $V_P$ and $V_{sd}$ as shown in Fig. 2e. By varying $V_{sd}$, we observe low conductance areas corresponding to the Coulomb blockade. The width of the Coulomb blockade regime is varied by $V_P$ (Fig. 2f), leading to the conductance map structure known as Coulomb diamonds. These observations demonstrate the electrostatic formation and control of the quantum dot. Here, we can estimate the charging energy ($E_C$) and the orbital level spacing ($\Delta\varepsilon$) from the nonuniform sizes between the Coulomb diamonds, as $E_C = 1.3$ meV and $\Delta\varepsilon = 0.3$ meV as indicated in Fig. 2e. $\Delta\varepsilon$ can also be estimated by analyzing the signal of cotunneling[49], and it is of the same order as the orbital energy mentioned above. In this experiment, we cannot unambiguously determine the absolute number of electrons in the dot because a negative $V_P$ lower than −5.0 V suppresses the current across the dot.

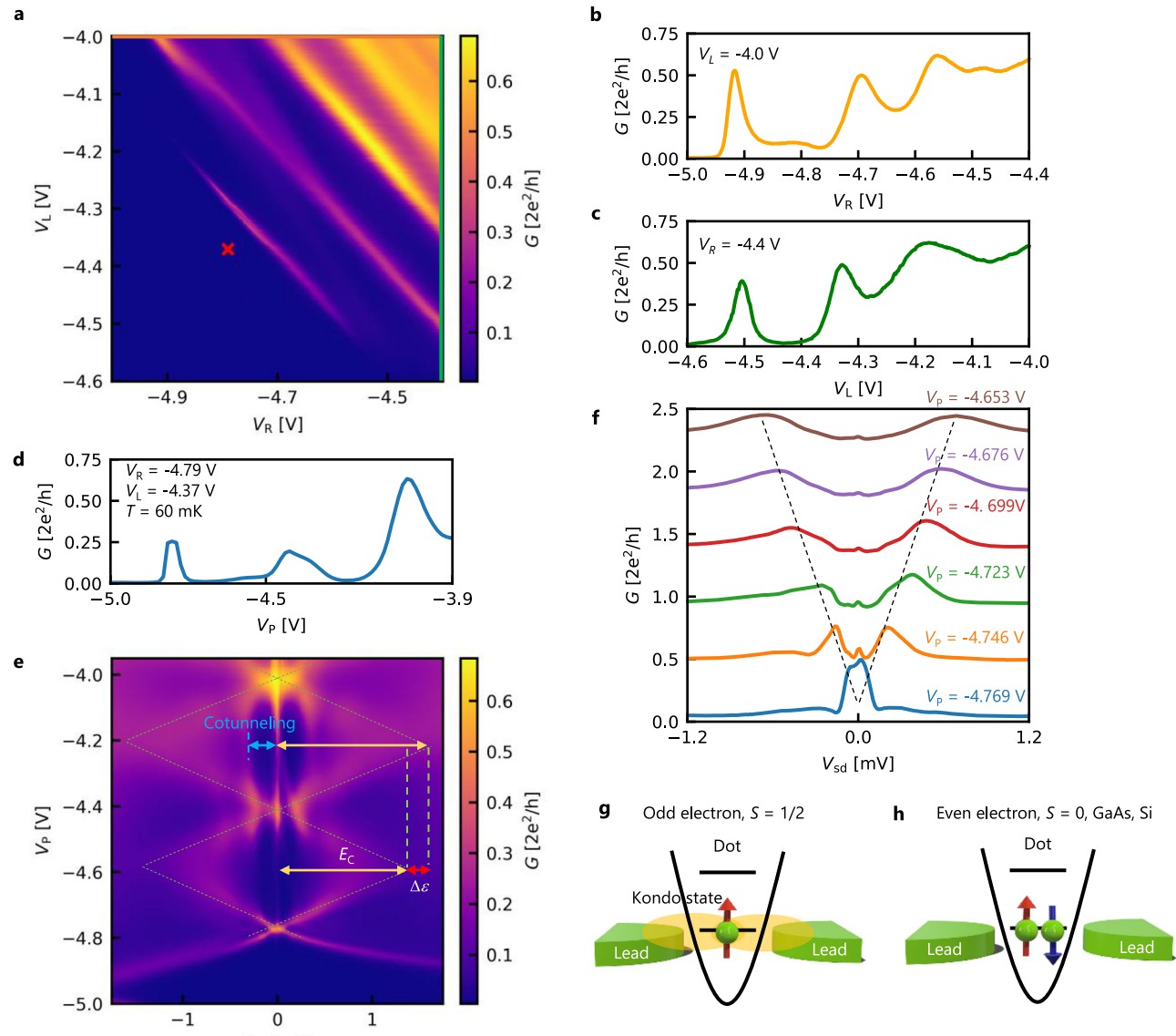

**Fig. 2 | Characteristics of the quantum dot. a** Conductance map measured as functions of $V_R$ and $V_L$ at $V_P = -5.0$ V. Coulomb oscillations appear when tunnel barriers are balanced to form a dot state. The red marker denotes the point of $V_L$ and $V_R$ used in (**e, f**). **b, c** Conductance measured as a function of $V_R$ and $V_L$ along the lines in (**a**), respectively. **d** Conductance measured as a function of $V_P$ with $V_L = -4.37$ V and $V_R = -4.79$ V. **e** Conductance map measured as functions of $V_P$ and $V_{sd}$. The dotted lines are the guides to the eyes, indicating the blockade regime. The charging energy ($E_C$) and the orbital level spacing ($\Delta\varepsilon$) are also indicated. The blue line shows the observed cotunneling signal. **f** Conductance measured as a function of $V_{sd}$ with changing $V_P$ from $-4.769$ to $-4.653$ V. Each trace is vertically shifted by 0.45 ($2e^2/h$) for clarity. **g, h** Schematic diagrams of spin filling and Kondo state in the cases of odd and even electrons.

## Kondo effect

Notably, in Fig. 2e, we notice distinct conductance peaks at $V_{sd} = 0$ V, reminiscent of the Kondo effect in the quantum dot[13,15,16]. The Kondo effect occurs when itinerant electrons screen the localized spins in the quantum dot, resulting in a coherent co-tunneling process[50]. Therefore, the number of electrons in the dot should be odd because an unpaired localized spin is necessary for the appearance of the Kondo effect(Fig. 2g). In contrast, in the presence of an even number of electrons, the Kondo effect is usually absent since the singlet $S = 0$ state is typically stable (Fig. 2h). Experimentally, however, we observe zero-bias peaks in the neighboring Coulomb diamonds, meaning that the Kondo effect manifests regardless of the even-odd electron parity in the case of our ZnO quantum dot, which will be further discussed later. Because the tunnel rate changes significantly with the change in plunger gate voltage, the coupling between the QD and the source-drain electrodes changes quickly from strong to weak, which makes it

hard to observe Coulomb diamonds in a wide range of gate voltages. We confirmed the absence of even-odd parity in the Kondo effect as observed in other devices(Supplementary Fig. 1).

To verify the Kondo effect, we measure the temperature dependence of the conductance as a function of $V_P$ as shown in Fig. 3a, at temperatures from 70 to 600 mK. The zero-bias conductance at $V_{sd} = 0$ V is suppressed in the Coulomb-blockaded regions with increasing temperature. This behavior is more evident in the conductance spectra as a function of $V_{sd}$ at the midpoint of the Coulomb-blockaded regions at $V_P = -5.04$, $-4.64$, and $-4.30$ V, denoted as $N-1$, $N$, and $N+1$, as shown in Fig. 3b, d, and f, respectively. The peak structure diminishes rapidly with increasing temperature, consistent with a characteristic feature of the Kondo effect.

To delve into the Kondo effect more quantitatively, we plot the temperature dependence of the zero-bias conductance peak value in Fig. 3c, e, and g. These plots exhibit a clear $\ln(T)$ dependence, a

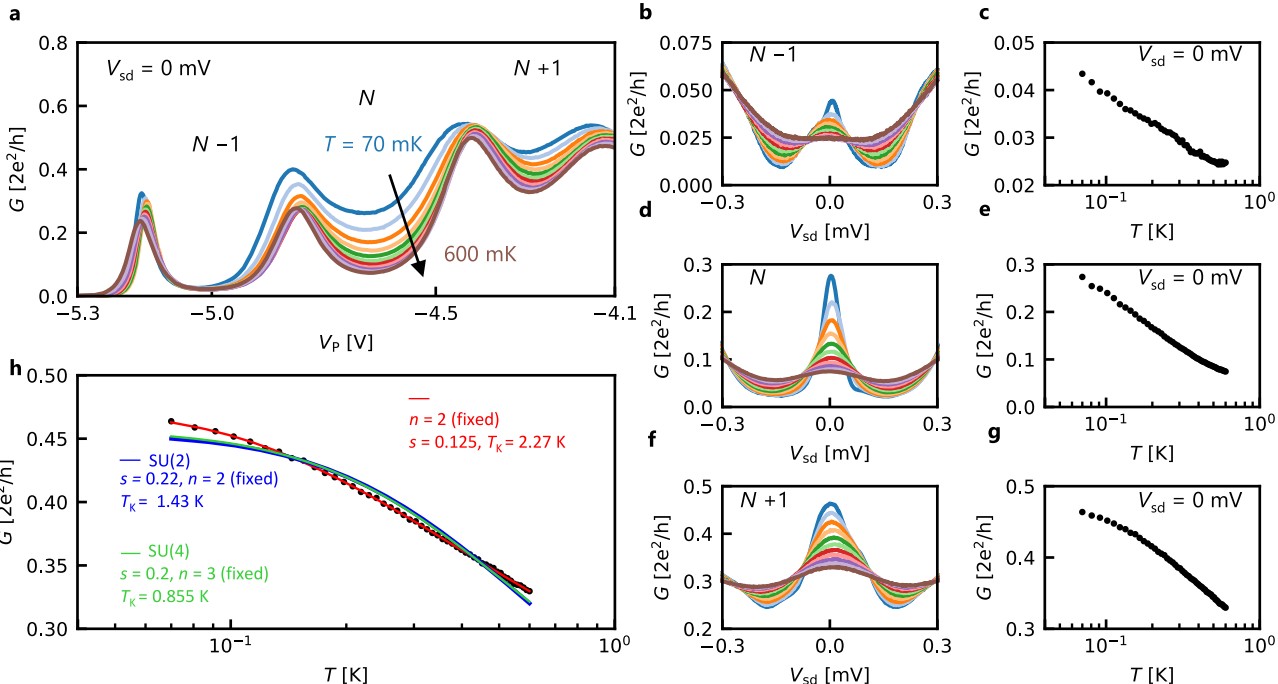

**Fig. 3 | Temperature dependence of the Kondo effect. a** Plunger gate voltage ($V_P$) dependence of conductance measured at $T$ = 70, 123, 176, 229, 282, 335, 388, 441, 494, 547, and 600 mK. **b, d, f** Source-drain voltage ($V_{sd}$) dependence of conductance measured at the same temperature range for $V_P$ = −5.04, −4.64, and −4.30 V, corresponding to the electron number of $N-1$, $N$, and $N+1$, respectively. **c, e, g** Temperature dependence of zero-bias peak conductance corresponding to the (**b, d, f**). **h** The fitting to the data shown in (**g**) using Eq. (2).

characteristic feature of the Kondo effect, which is known to be pronounced around the temperature range of $0.1T_K < T < T_K$ ($T_K$: Kondo temperature). Outside of this temperature range, according to the linear response theory, the conductance ($G$) follows a temperature dependence of $\sim 1/\ln^2(T/T_K)$ at $T \gg T_K$ and asymptotically approaches $G_0$ with a Fermi liquid temperature dependence of $\sim -(T/T_K)^2$ at $T \ll T_K$[51]. Here, $G_0$ is the conductance in the low-temperature limit and is expressed as

$$G_0 = G_s \frac{4|t_L^2 t_R^2|}{(\mu - \varepsilon_0)^2 + (|t_L^2| + |t_R^2|)^2},\tag{1}$$

where $t_L$ and $t_R$ are the transmission coefficients from the dot to the left and right reservoir, respectively, and $G_s$ is the quantum of conductance $2e^2/h$. $G_0$ is maximum when $t_L = t_R$[51]. $\mu$ and $\varepsilon_0$ show Fermi energy and the single-particle energy level in the quantum dot. By fitting the Coulomb peak at $T$ = 70 mK in Fig. 3a, $|t_L^2|$, $|t_R^2|$ are estimated to be 0.13 meV and 0.68 meV, respectively. $T_K$ is given by $T_K = \left(\sqrt{\Gamma E_C}/2\right)\exp\{\pi\varepsilon_0(\varepsilon_0 + E_C)/\Gamma E_C\}$[16], where $\Gamma = \Gamma_L + \Gamma_R = |t_L^2| + |t_R^2|$, and $T_K$ becomes 1.7 K. This also supports that the system is in the Kondo regime. Moreover, we calculated $T_K$ by using full width at half maximum (FWHM) of the zero-bias peak at $T$ = 70 mK in Fig. 3f. In earlier studies, the relation $T_K = e \cdot \text{FWHM}/k_B$ was used to estimate $T_K$[16]. By using this relation $T_K$ was calculated to be 1.7 K, which is consistent with the calculation by Eq. (1) discussed above. For fitting the experimental temperature dependence, it is convenient to use the following empirical form[14,22]

$$G(T) = G_0 \left(\frac{T_K'^2}{T^2 + T_K'^2}\right)^s,\tag{2}$$

where

$$T_K' = \frac{T_K}{\left(2^{1/s} - 1\right)^{1/n}}.\tag{3}$$

The fitting using Eq. (2) to the case of $N+1$ is shown in Fig. 3h, yielding $G_0 = 0.475(2e^2/h)$, $s = 0.125$, $n = 2$, $T_K = 2.27$ K. These fitting parameters provide valuable insights into the peculiar features of the Kondo effect in ZnO. In the simplest case of nondegenerate $S = 1/2$ (SU(2)), we would expect the exponents around $s = 0.22$ and $n = 2$[14,52]. This discrepancy in the exponents is unexpected since ZnO has a nondegenerate single electron band, similar to GaAs, and therefore SU(2) symmetry would be expected. Even assuming the presence of doubly degenerate orbitals (SU(4)) as in carbon nanotube or graphene[18,21,22], a renormalization group approach predicts $s = 0.20$ and $n = 3$[52], inconsistent with the present case of ZnO. The fittings constraining $s = 0.22$ or $s = 0.20$ fail to explain the observed temperature dependence as shown in Fig. 3h, ruling out the possibilities of the SU(2) and SU(4) Kondo effects with $S = 1/2$.

In the case of ZnO, we need to consider an alternative perspective on the peculiar Kondo effect. ZnO is recognized for its relatively strong electron correlation because of a small dielectric constant ($\epsilon = 8.3\epsilon_0$, $\epsilon_0$: vacuum permittivity) compared with conventional semiconductors such as GaAs and Si ($\epsilon(\text{GaAs}) = 12.9\epsilon_0$, $\epsilon(\text{Si}) = 11.7\epsilon_0$)[47,53]. Together with a large effective mass of $m = 0.3m_0$ ($m_0$: mass of a bare electron), which results in relatively small orbital energy splitting, this property has led to many unconventional phenomena in the two-dimensional[39–43] and one-dimensional[48] electrons in ZnO. Consequently, we could anticipate an unconventional phenomenon stemming from the correlation effect in the ZnO quantum dot as well. One possible scenario is that Hund's coupling energy may exceed the orbital separation energy, $\Delta\varepsilon$, thereby stabilizing the $S \geq 1/2$ Kondo state, regardless of the even-odd electron filling. In fact, a numerical calculation for the $S = 1$ triplet Kondo state in ref. 54 demonstrates that Eq. (2) best fits the temperature dependence with $s \approx 0.15$, assuming $n = 2$, close to the fitting parameters in our data. Future further measurements in various conditions and comparison of the results with numerical renormalization group (NRG) calculations[55–57] will contribute to understanding the detailed temperature dependence.

The triplet $S = 1$ Kondo effect in the semiconductor quantum dot has been discussed, but the Kondo temperature is predicted to be

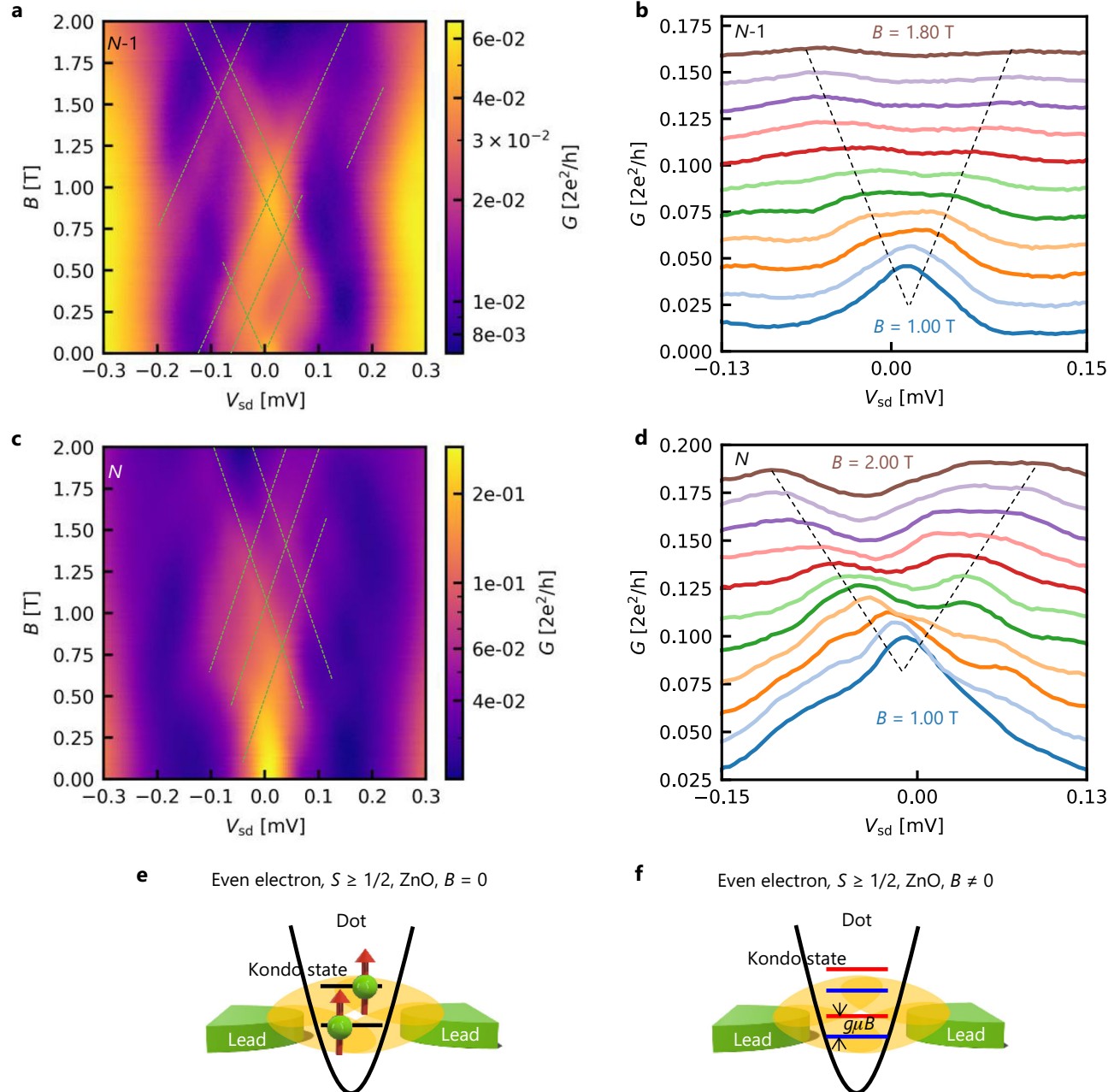

**Fig. 4 | Magnetic field dependence of the Kondo effect. a, c** Conductance map as functions of magnetic field ($B$) and source-drain voltage ($V_{sd}$) for $N-1$ (**a**) and $N$ (**b**). Multiple peaks exist as indicated by the dotted lines for the eye guide. **b, d** Conductance measured as a function of $V_{sd}$ with changing $B$ by 0.08 T setup (0.1 T in **d**). Each trace is vertically shifted by 0.015 ($2e^2/h$). **e, f** Schematic diagrams of spin filling and Kondo state in the case of ZnO without and with a magnetic field.

several orders of magnitude lower than that for the $S = 1/2$[55,58]. Instead, the even-electron Kondo effect is realized at the singlet-triplet level degeneracy under a magnetic field[17,20]. To access this possibility in our case, we measure the magnetic field dependence of the conductance for the cases of $N-1$ and $N$ as shown in Fig. 4a–b and c–d, respectively. In both cases, we observe several field-dependent peak structures in the Coulomb-blockaded region. The energy scales of the magnetic field dependence are estimated as 0.20 mV/T for $N-1$ and 0.28 mV/T for $N$. This energy scale roughly aligns with the effect of Zeeman splitting for the $S = 1/2$ Kondo state, $2g\mu_B = 0.22$ meV/T, with $g = 1.94$ for the ZnO 2DEG[59]. However, the Kondo peak at the singlet-triplet degeneracy is known to be much more sensitive to the magnetic field than the Zeeman splitting effect as observed in ref. 17,20, unlikely to be the origin of our observation. The $S = 1$ triplet Kondo effect suggested above is not plausible either because additional peak splitting

equivalent to $4g\mu_B B$ corresponding to $|T^-\rangle \rightarrow |T^+\rangle$ is expected in addition to the splitting of $2g\mu_B B$ corresponding to $|T^-\rangle \rightarrow |T^0\rangle$. However, we cannot completely rule out this possibility if the two-spin flip process ($|T^-\rangle \rightarrow |T^+\rangle$) is too weak to observe as in the case of bilayer graphene[22].

Having considered these possibilities, we also propose another mechanism of the observed even-odd independent Kondo effect that involves multiple orbitals strongly hybridized with each other as indicated by the complex peak structures in Fig. 4. Electrons occupy higher orbitals with remaining unpaired spins ($S \geq 1/2$) instead of forming a singlet state due to intra-dot correlations. In this case, each localized spin may be independently coupled to surrounding electrons, resulting in multiple scales of the Kondo temperatures (Fig. 4e, f). This makes the interpretation by fitting with Eq. (2) inappropriate. We note that a similar discussion has been presented in ref. 19

regarding the Kondo effect in the GaAs quantum dot, where even-odd behavior is absent when energy separation $\Delta\varepsilon$ is smaller than the energy scales of temperature, $k_BT$, or energy broadening due to tunnel coupling, $\Gamma$. However, our observation differs from this case, having a relatively large energy separation of 0.3 meV, corresponding to a temperature of 3.5 K. Moreover, this breakdown of even-odd effects is commonly observed in different devices in the case of ZnO (See the Supplementary Information). Nevertheless, we cannot completely rule out the possibility of the singlet-triplet or $S = 1$ Kondo effect because of the limitation of the detailed state analysis. For a more detailed understanding, the energy spectrum should be investigated over a broader range of electron filling, accompanied by a numerical calculation using the parameters specific to ZnO, which remains to be investigated in the future. The observation of quantum dots in ZnO that can realize clean and correlated electron systems, and the characteristic Kondo effect reflecting the properties, is expected to provide controllability for future quantum technologies utilizing electron correlation different from that of GaAs and Si.

In this study, we have successfully demonstrated the electrostatically defined quantum dot device using high-quality ZnO heterostructures. Transport measurement through the dot exhibits the clear Coulomb peaks and Coulomb diamonds, confirming the formation and control of the quantized states. Additionally, zero-bias peaks, indicative of the Kondo effect, are observed in the Coulomb diamond, which unexpectedly appears independent of the electron number parity. Through the measurement of temperature and magnetic field dependences of the Kondo resonance peaks, we suggested that multiple orbitals in the dot may be involved due to strong electron interaction. Our results open new avenues for exploring new physics and applications of quantum dots, leveraging the distinctive properties of ZnO, such as a simple single electron band, a relatively weak spin-orbit interaction, low-density nuclear spins, and strong correlation effects.

## Methods

### Sample fabrication

(Mg,Zn)O/ZnO heterostructures are grown on Zn-polar ZnO (0001) substrates at 750 °C by molecular beam epitaxy using distilled pure ozone as an oxygen source. Mg content in the heterostructure used in this study is about 2.5 %. The electron density and mobility are measured by the Hall effect as $n = 4.9 \times 10^{11}\,\mathrm{cm^{-2}}$ and $\mu = 170{,}000\,\mathrm{cm^2\,V^{-1}\,s^{-1}}$, respectively, at 1.8 K. The Ti/Au ohmic electrodes are fabricated by photolithography and lift-off process The AlO$_x$ gate insulator is deposited by atomic layer deposition. The standard electron-beam lithography and lift-off techniques are used to form the Ti/Au top split gate electrodes.

### Transport measurement

The transport properties are measured in a dilution refrigerator equipped with a superconducting magnet. The base temperature is 56 mK. In the temperature-controlled measurements, a heater in the refrigerator is controlled by a PID controller. The gate voltages are supplied by DC voltage sources, and the values are optimized to form the confinement potential of the quantum dot. The conductance of the device is measured using a lock-in amplifier with an excitation frequency of 210 Hz and a voltage of 6 μV. The current from the device is amplified by a current amplifier that converts the current to voltage, and the voltage is supplied to the lock-in amplifier.

## Data availability

The data that support the findings of this study are available in the article and its Supplementary Information. Additional data related to this paper may be requested from the authors.

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

## Acknowledgements

The authors thank M. Eto, R. Sakano, W. Izumida, M. Takeuchi, A. Kurita, RIEC Fundamental Technology Center, and the Laboratory for Nanoelectronics and Spintronics for fruitful discussions and technical support. Part of this work was supported by MEXT Leading Initiative for Excellent Young Researchers, Grants-in-Aid for Scientific Research (21K18592, 22H04958, 23H01789, 23H04490), Tanigawa Foundation Research Grant, Maekawa Foundation Research Grant, The Foundation for Technology Promotion of Electronic Circuit Board, Iketani Science and Technology Foundation Research Grant, The Ebara Hatakeyama Memorial Foundation Research Grant, FRiD Tohoku University, and "Advanced Research Infrastructure for Materials and Nanotechnology in Japan (ARIM)" of the Ministry of Education, Culture, Sports, Science and Technology (MEXT) (Proposal Number JPMXP1223NM5159). AIMR and MANA are supported by World Premier International Research Center Initiative (WPI), MEXT, Japan.

## Author contributions

Y.K. and T.O. conceived the ideas. K.N., Y.K., and T.O. led the experiments. Y.K., T.K., A.T., M.K., and T.O. fabricated the samples. K.N., K.M., T.K., Y.F., and T.O. performed the transport measurements. K.N., Y.K., K.M., Y.F., and T.O. analyzed the data and all the authors discussed the results. K.N., Y.K., and T.O. wrote the paper with inputs and comments from all authors. T.O. supervised the project.

## Competing interests

The authors declare no competing interests.
