## [Transparent Peer Review file · Nature Communications]

Parity-independent Kondo effect of correlated electrons in electrostatically defined ZnO quantum dots

Corresponding Author: Professor Tomohiro Otsuka

Version 0:

Reviewer comments:

Reviewer #1

(Remarks to the Author)

The authors present experimental data on electronic transport through a gate-defined quantum dot in ZnO. The authors demonstrate control over the electron number in the quantum dot (although without knowing the absolute number of electrons in the dot) by performing bias spectroscopy and measuring Coulomb diamonds. In particular, the authors investigate conductance peaks around zero bias for consecutive Coulomb diamonds. Temperature dependence of these zero-bias peaks coincide with the physics of the Kondo effect, where coupling between the leads and the quantum dot is sufficiently large and leads to cotunneling processes depending on the spin state of the dot. Furthermore, the authors perform a fitting analysis of the temperature dependence of the Kondo peaks based on empirical equations. This fitting differs from the expected result for a system with SU(2) symmetry such as ZnO. Finally, the authors rule out singlet-triplet degeneracy (as observed by the Kouwenhoven group in GaAs) and small orbital energy in the dot (as observed by Von Klizing's group in GaAs) and explain the deviation from the usual even-odd behavior of the Kondo effects by invoking stronger electron correlations in ZnO than in GaAs.

The structure of the manuscript is well presented. The English is clear, but not polished (probably due to language barriers), but that can easily be fixed. This device is, to my knowledge, the first experimental realization of an electrostatic ZnO quantum dot, which is a good technical achievement. The stability diagram of the quantum dot as well as the temperature dependence of the Kondo peaks is clear and convincing. I do however have some issue with the clarity of figures 2e (Coulomb diamonds) and figures 4a and 4c (magnetic field dependence of the Kondo peaks). This causes some problem with the message the authors convey. They use Coulomb diamonds to extract the orbital energy of the dot and therefore rule out the argument by which small orbital energies and Hund's rules could explain the break-down of the usual even-odd behaviour of the Kondo effect. As explained in point 5 below, I don't think the analysis to extract the orbital energy from the Coulomb diamonds is strong enough. The authors also rule out the singlet-triplet degeneracy at a finite magnetic field to explain even occupation Kondo effect, based on the dependence of the Kondo peaks as a function of magnetic field (figs 4a,c), which in my opinion are not clear enough.

For these reasons, I don't recommend publication of this manuscript in Nature Communications in its present form. However, if the authors could find more convincing data or analysis to rule out already demonstrated explanations for the observation of the Kondo effect at even number of electrons, I think that would warrant publication in the journal, specially considering that this is the first gated quantum dot in ZnO. Additionally, I suggest the authors address the points which I list below.

- 1- Page 3, Paragraph 2: can the authors define the electron interaction parameter r_s ?
- 2- Page 3, Paragraph 2: can the authors state what the band gap in ZnO is?
- 3- Page 3, Paragraph 2: what is the implication of a 'single electron pocket' in the Fermi surface for long spin coherence? Do the authors mean that there are no other degeneracies such as multiple valleys?
- 4- Page 3, Paragraph 2: Weak spin orbit interaction is a double edge sword in this case: yes, the SOI can lead to decoherence, but in many semiconducting spin qubit implementations the SOI is exploited to electrically manipulate the spin. This is a much more practical way than employing oscillating magnetic fields to coherently drive a spin qubit in a quantum dot.
- 5- Page 3, Paragraph 2: Can the authors state what is the total nuclear spin for ZnO?
- 6- Fig1 caption: 'Gate electrodes are formed on the AlOx gate insulator'... probably the authors mean 'fabricated on top of

the...'

7- Page 5, paragraph 2: The authors claim that they estimate charging and orbital energies by analyzing the slopes of the Coulomb diamonds in fig.2e. I find that the data in this figure is convincing to demonstrate Coulomb blockade and also the zero-bias conductance peak related to the Kondo effect, but it is very weak to differentiate addition and orbital energies. Unless they have a more solid analysis to extract the orbital energy, I would avoid this argument.

8- Can the authors extract the tunnel rate to the reservoirs and compare it to the energy scales of the system to numerically determine if they are sufficiently large to be in the Kondo regime?

9- Eq. 1: the authors write 'G_S is the maximum value when $t_L = t_R$ and is equal to $2e^2/h$ in the case of $S = 1/2$ Kondo state'. This is confusing. Could the authors just say: 'where G_S is the quantum of conductance $2e^2/h$. G_0 is maximum when $t_L = t_R$ '?

10- The authors extract T_K as a parameter from eqs. 2 and 3. Could they extract T_K from the FWHM of the zero-bias peak at base temperature and compare the values obtained with eqs. 2 and 3?

11- Page 11, paragraph 2: The authors eliminate small orbital energies compared to $k_B T$ as an explanation for the breakdown of the even-odd behavior of the Kondo effect. This is well explained in Von Klitzing's paper as pointed out by the authors. As I explained in point 7, the analysis to extract orbital energy seems weak, so unless they can prove me wrong, they can't rule out the explanation by which the addition of electrons follow Hund's rule and therefore parallel spin configurations are favorable due to exchange interaction, deviating from the usual even-odd behavior.

Reviewer #2

(Remarks to the Author)

The paper by Noro et al. introduces an electrostatic approach to forming quantum dots in ZnO heterostructures, as previously has been demonstrated in, e.g., GaAs and Si systems. In electron transport measurements, manifestations of the Kondo effect are observed, however without clear even-odd electron number parity. The authors suggest that their work not only sheds light on correlated electron physics in quantum dots, but they also propose potential applications in quantum devices, capitalizing on ZnO's unique properties.

The work represents a solid and very interesting study of the Kondo effect in quantum dots. The claimed impact for quantum devices in general, and spin qubits in particular could be better explained and substantiated in the paper.

I wonder why only two full Coulomb diamonds are studied in Fig. 2e, which in my opinion makes it difficult to make strong and general statements about odd-even asymmetry in this system. Do the authors have data showing more consecutive Coulomb diamonds, which could for example be included in the Supplementary Material? In how many QD devices similar effects have been observed?

In Fig. 3h, could the authors compare their results with the exact NRG results for $S=1/2$ and $S=1$ in a G/G_0 vs T/T_K plot? The authors should also compare their results with the predictions for a two-stage Kondo effect for $S \geq 1$, as described in Phys. Rev. Lett. 88, 126803 (2002).

Minor points:

The authors should define the electron interaction parameter r_S on p3.

p.9 typo: singlet-triplet

p.10 typo in caption Fig. 4: without

Version 1:

Reviewer comments:

Reviewer #1

(Remarks to the Author)

I have carefully reviewed the response from the authors. They have addressed adequately all my comments. In particular, establishing the energies using the co-tunneling signal corroborates their previous interpretation.

I therefore recommend publication in Nature Communications of this improved version of the manuscript.

Reviewer #2

(Remarks to the Author)

The authors have addressed the points I raised about the original version of the manuscript.

(1) Multiple Coulomb diamonds and observation in other devices

The authors explain convincingly that the coupling between the QD and the source-drain electrodes changes quickly from strong to weak when sweeping the plunger gate voltage. I understand that it is therefore difficult to study several consecutive Coulomb diamonds in a single device.

The authors do supply data of two other devices in the Supplementary Information. The data are not very clear (especially the resolution of Fig. S1b is rather low), but there seem to be zero-bias peaks in neighbouring Coulomb diamonds. Based on the provided data, it actually looks like there is always a zero-bias peak once the conductance (coupling) is high enough.

(2) Comparison with NRG results/two-stage Kondo effect

The authors compared some of their results with the NRG results for $S = 1/2$ and $S = 1$ in a G/G_0 vs T/TK plot. However, it is not clear to me what Coulomb diamond they used for this analysis, and whether they found the same results in other (neighbouring) diamonds. This is a critical omission.

The presented data can clearly not be fitted by NRG results for $S = 1/2$ nor $S = 1$. I therefore do not understand the remark "Nevertheless, we cannot completely rule out the possibility of the singlet-triplet or $S = 1$ Kondo effect because of the limitation of the detailed state analysis." (line 199-200).

Indeed, I would argue that based on the mismatch with the NRG results, a more in-depth analysis is required where one would systematically investigate the role of (1) multi-channel Kondo effect, (2) spin-orbit coupling, (3) anisotropic Kondo effect, (4) higher spin states, (5) mixed-valence regime, (6) decoherence effects, (7) non-Fermi-liquid behaviour. The authors adequately address the two-stage Kondo effect question.

I agree with the authors that "Future further measurements in various conditions and comparison of the results with numerical renormalization group (NRG) calculations[55–57] will contribute to understanding the detailed temperature dependence." (line 168-170). Given the amount of open questions, I find the claim for "parity-independent Kondo effect of correlated electrons" slightly premature and not very well substantiated. Although the work is definitely interesting, overall, the manuscript appears to me more as "work in progress".

The results will be definitely of interest to a certain readership. Whether the manuscript in its present form is mature enough to warrant publication in Nature Communications is a judgement I leave to the Editor.

Response to Referee #1

We are grateful to the referee for careful and critical reading of our manuscript. We have revised our manuscript according to the comments. Our responses to the comments are described below.

Summary comment. *The authors present experimental data on electronic transport through a gate-defined quantum dot in ZnO. The authors demonstrate control over the electron number in the quantum dot (although without knowing the absolute number of electrons in the dot) by performing bias spectroscopy and measuring Coulomb diamonds. In particular, the authors investigate conductance peaks around zero bias for consecutive Coulomb diamonds. Temperature dependence of these zero-bias peaks coincide with the physics of the Kondo effect, where coupling between the leads and the quantum dot is sufficiently large and leads to cotunneling processes depending on the spin state of the dot. Furthermore, the authors perform a fitting analysis of the temperature dependence of the Kondo peaks based on empirical equations. This fitting differs from the expected result for a system with SU(2) symmetry such as ZnO. Finally, the authors rule out singlet-triplet degeneracy (as observed by the Kouwenhoven group in GaAs) and small orbital energy in the dot (as observed by Von Klizing's group in GaAs) and explain the deviation from the usual even-odd behavior of the Kondo effects by invoking stronger electron correlations in ZnO than in GaAs.*

The structure of the manuscript is well presented. The english is clear, but not polished (probably due to language barriers), but that can easily be fixed. This device is, to my knowledge, the first experimental realization of an electrostatic ZnO quantum dot, which is a good technical achievement. The stability diagram of the quantum dot as well as the temperature dependence of the Kondo peaks is clear and convincing.

Reply: We are pleased to learn that the reviewer has recognized the novelty and importance of our work. We also thank the reviewer for summarizing the critical points of our work.

I do however have some issue with the clarity of figures 2e (Coulomb diamonds) and figures 4a and 4c (magnetic field dependence of the Kondo peaks). This causes some problem with the message the authors convey. They use Coulomb diamonds to extract the orbital energy of the dot and therefore rule out the argument by which small orbital energies and Hund's rules could explain the break-down of the usual even-odd behaviour of the Kondo effect. As explained in point 5 below, I don't think the analysis to extract the orbital energy from the Coulomb diamonds is strong enough. The authors also rule out the singlet-triplet degeneracy at a finite magnetic field to explain even occupation Kondo effect, based on the dependence of the Kondo peaks as a function of magnetic field (figs 4a,c), which in my opinion are not clear enough.

Reply: We thank the referee for this comment. Regarding the orbital energy, we additionally analyzed the energy from the cotunneling signal. The details are explained in the reply to Comment 7.

We also considered and agree with the referee about the possibility of the singlet-triplet. We have modified the expression. The details are explained in the reply to Comment 11.

Comment 1. Page 3, Paragraph 2: can the authors define the electron interaction parameter r_s ?

Reply: Thank you for this comment. The interaction parameter r_s is defined as the ratio of the Coulomb energy to the Kinetic energy, and is defined as

$$r_s = m^* e^2 / 4\pi \hbar^2 \varepsilon \sqrt{n\pi},$$

where m^* is the effective mass, e is the elementary electric charge, \hbar is the Planck constant divided by 2π , ε is the dielectric constant, and n is the sheet carrier density. Accordingly, we have added the definition of the r_s as follows.

Changed(p. 2, l. 41)

“Figure 1a summarizes the feature of ZnO compared to other semiconductor materials in terms of the electron interaction parameter r_s and the transport scattering time τ .”

is now

" Figure 1a summarizes the feature of ZnO compared to other semiconductor materials in terms of the electron interaction parameter r_s and the transport scattering time τ , where r_s is defined as the ratio of the Coulomb energy to the Kinetic energy, and is expressed as $r_s = m^* e^2 / 4\pi \hbar^2 \varepsilon \sqrt{n\pi}$ (m^* : effective mass, e : elementary electric charge, \hbar : Planck constant divided by 2π , ε : dielectric constant, n : sheet carrier density).

Comment 2. Page 3, Paragraph 2: can the authors state what the band gap in ZnO is?

Reply: We appreciate the referee's comment. The band gap of ZnO is 3.37 eV. We have added the value of the band gap of ZnO and a corresponding reference as Ref. [44].

Changes(p. 3, l. 47)

“In addition to the correlation effect, ZnO stands out as a unique material compared to conventional semiconductors, characterized by its large band gap with a single electron pocket, weak spin-orbit interaction, and low-density nuclear spins.”

is now

“In addition to the correlation effect, ZnO stands out as a unique material compared to conventional

semiconductors, characterized by its large band gap ($E_g = 3.37$ eV)[44] with a single electron pocket preventing intervalley carrier scattering, weak spin-orbit interaction, and low-density nuclear spins (^{67}Zn (4% natural abundance) has a $I = 5/2$ nuclear spin, and ^{17}O (0.04% natural abundance) has a $I = 5/2$ nuclear spin, while other Zn and O isotopes show zero nuclear spin states.).”

Added (Reference)

[44]Özgür, Ü. *et al*, *J. Appl. Phys.* **98**, 041301 (2005).

Comment 3. *Page 3, Paragraph 2: what is the implication of a ‘single electron pocket’ in the fermi surface for long spin coherence? Do the authors mean that there are no other degeneracies such as multiple valleys?*

Reply: Thank you for providing this comment. As pointed out by the referee, “single electron pocket” means that ZnO is not a multi-valley system unlike Si. The degenerate conduction bands are known to be a source of electron relaxation via intervalley scattering. Therefore, a single valley system is more robust against carrier relaxation.

Changes(p. 3, l. 47)

“In addition to the correlation effect, ZnO stands out as a unique material compared to conventional semiconductors, characterized by its large band gap with a single electron pocket, weak spin-orbit interaction, and low-density nuclear spins.”

is now

“In addition to the correlation effect, ZnO stands out as a unique material compared to conventional semiconductors, characterized by its large band gap ($E_g = 3.37$ eV)[44] with a single electron pocket preventing intervalley carrier scattering, weak spin-orbit interaction, and low-density nuclear spins (^{67}Zn (4% natural abundance) has a $I = 5/2$ nuclear spin, and ^{17}O (0.04% natural abundance) has a $I = 5/2$ nuclear spin, while other Zn and O isotopes show zero nuclear spin states.).”

Comment 4. *Page 3, Paragraph 2: Weak spin orbit interaction is a double edge sword in this case: yes, the SOI can lead to decoherence, but in many semiconducting spin qubit implementations the SOI is exploited to electrically manipulate the spin. This is a much more practical way than employing oscillating magnetic fields to coherently drive a spin qubit in a quantum dot.*

Reply: We thank the referee for this comment. As pointed out by the referee, the spin-orbit interaction can be used for spin manipulation. But it also causes decoherence and the appropriate control of the interaction is crucial. In Si spin qubits, introducing controllable effective spin

interaction induced by micro-magnets in small spin-orbit materials are widely used [28]. Because of the small spin-orbit interaction in ZnO, the same approach can be employed. Following the referee's comment, we have added the following sentences in the text.

Added(p.3, l. 53)

Although spin-orbit interaction can be used for spin manipulation, it also causes decoherence and the appropriate control of the interaction is crucial. In Si spin qubits, introducing controllable effective spin interaction induced by micro-magnets in small spin-orbit materials are widely used[28]. Because of the small spin-orbit interaction in ZnO, the same approach can be employed.

Comment 5. Page 3, Paragraph 2: Can the authors state what is the total nuclear spin for ZnO?

Reply: Thank you for this comment. The natural abundance and nuclear spins of Zn and O isotopes are shown in the following table. ^{67}Zn has a nuclear spin of $I = 5/2$ (4% natural abundance), and ^{17}O has a nuclear spin of $I = 5/2$ (0.04 % natural abundance), while other Zn and O isotopes do not have nuclear spins.

Table: Natural abundance of Zn and O isotopes and their nuclear spins.

Isotope	Nuclear spin	Natural abundance
Zn ⁶⁴	0	0.48
Zn ⁶⁶	0	0.28
Zn ⁶⁷	5/2	0.041
Zn ⁶⁸	0	0.19
Zn ⁷⁰	0	0.0063

Isotope	Nuclear spin	Natural abundance
O ¹⁶	0	0.998
O ¹⁷	5/2	0.0004
O ¹⁸	0	0.002

Changes(p. 3, l. 47)

“In addition to the correlation effect, ZnO stands out as a unique material compared to conventional semiconductors, characterized by its large band gap with a single electron pocket, weak spin-orbit interaction, and low-density nuclear spins.”

is now

“In addition to the correlation effect, ZnO stands out as a unique material compared to conventional

semiconductors, characterized by its large band gap ($E_g = 3.37$ eV)[44] with a single electron pocket preventing intervalley carrier scattering”, weak spin-orbit interaction, and low-density nuclear spins (^{67}Zn (4% natural abundance) has a $I = 5/2$ nuclear spin, and ^{17}O (0.04% natural abundance) has a $I = 5/2$ nuclear spin, while other Zn and O isotopes show zero nuclear spin states.).”

Comment 6. *Fig1 caption: ‘Gate electrodes are formed on the AlO_x gate insulator’... probably the authors mean ‘fabricated on top of the...’*

Reply: We appreciate the referee's comment. We have corrected the caption of Fig. 1 as “Gate electrodes are fabricated on top of the AlO_x gate insulator”.

Changes(Fig. 1 caption)

" Gate electrodes are formed on the AlO_x gate insulator."

is now

" Gate electrodes are fabricated on top of the AlO_x gate insulator."

Comment 7. *Page 5, paragraph 2: The authors claim that they estimate charging and orbital energies by analyzing the slopes of the Coulomb diamonds in fig.2e. I find that the data in this figure is convincing to demonstrate Coulomb blockade and also the zero-bias conductance peak related to the Kondo effect, but it is very weak to differentiate addition and orbital energies. Unless they have a more solid analysis to extract the orbital energy, I would avoid this argument.*

Reply: Thank you for providing this comment. We additionally analyzed the energy from the cotunneling signal. The orbital energy can be estimated from the cotunneling features observed in the Coulomb diamonds [De Franceschi, S. *et al*, PRL. **85**, 878-881(2001)] as shown in the following figure. The orbital energies $\Delta\epsilon$ estimated in this way are of the same order as those estimated from the sizes of the Coulomb diamonds. Therefore, we consider the values seem to be appropriate. We have added this information to the main text.

FIG. : Analysing the cotunneling signal to estimate orbital energy

Added(p. 6, l. 100)

$\Delta\varepsilon$ can also be estimated by analyzing the signal of cotunneling[49], and it is of the same order as the orbital energy mentioned above.

Added(Figure.2 caption)

The blue line shows the observed cotunneling signal .

Changes(Figure.2e)

Comment 8. Can the authors extract the tunnel rate to the reservoirs and compare it to the energy scales of the system to numerically determine if they are sufficiently large to be in the Kondo regime?

Reply: We thank the referee for this comment. We have conducted the suggested analysis. We estimated $\Gamma (= \Gamma_L + \Gamma_R)$ and attempted to calculate Kondo temperature by fitting the Coulomb peak between electron numbers N and $N+1$ at $T = 56$ mK in the Fig.3(a) in the manuscript.

Γ_L and Γ_R can be estimated by fitting the Coulomb peak with

$$G = \frac{2e^2}{h} \frac{4\Gamma_L\Gamma_R}{(\mu - \varepsilon_0)^2 + (\Gamma_L + \Gamma_R)^2} = \frac{2e^2}{h} \frac{4\Gamma_L\Gamma_R}{[\alpha(V_{\text{peak}} - V_P)]^2 + (\Gamma_L + \Gamma_R)^2}. \quad (1)$$

Here, V_{peak} is V_P where the conductance of Coulomb peak becomes maximum value, thus the energy level within the quantum dot is aligned with the Fermi level at V_{peak} . Now, $V_{\text{peak}} = -4.44$ V. And α is α -factor. α is given by

$$\alpha = \frac{C_g}{C} = \frac{e/\Delta V_g}{e^2/E_C} = 4.3 \times 10^{-3}. \quad (2)$$

Γ_L and Γ_R are roughly estimated to be 0.13 ± 0.0029 meV and 0.68 ± 0.021 meV respectively by

using these values and equations. Thus Γ can be calculated to be $\Gamma = 0.81$ meV.

$$\Gamma = 0.13 + 0.68 = 0.81 \text{ meV}$$

The result of fitting Coulomb peak with eq.(1).

The previous study[van der Wiel *et al*, Science **289**, 2105-2108(2000).] shows that T_K is calculated by

$$T_K = \frac{1}{k_B} \frac{\sqrt{\Gamma E_C}}{2} \exp \frac{\pi \varepsilon_0 (\varepsilon_0 - E_C)}{\Gamma E_C} \quad (3)$$

Here, E_C is charging energy and estimated to be 1.3 meV, and ε_0 is the energy difference between Fermi level and single-particle level in the quantum dot. Setting $\varepsilon_0 = -0.5E_C$, we obtained $T_K = 1.7$ K.

This estimated T_K is larger than the temperature of experiment system, and thus tunnel rate Γ thought to be large enough to observe Kondo Effect. To clarify these points, we have changed and added the following sentences in the text.

Changes(Eq.1)

$$G = G_S \frac{4|t_L^2 t_R^2|}{(|t_L^2| + |t_R^2|)^2}$$

is now

$$G = G_S \frac{4|t_L^2 t_R^2|}{(\mu - \varepsilon_0)^2 + (|t_L^2| + |t_R^2|)^2}$$

Added(p. 8, l.135)

μ and ε_0 show Fermi energy and the single-particle energy level in the quantum dot. By fitting the

Coulomb peak at $T = 56$ mK in Fig.3a, $|t_L^2|$ and $|t_R^2|$ are estimated to be 0.13 meV and 0.68 meV, respectively. T_K is given by $T_K = (\sqrt{\Gamma E_C}/2) \exp\{\pi\varepsilon_0(\varepsilon_0 - E_C)/\Gamma E_C\}$ [16], where $\Gamma = \Gamma_L + \Gamma_R = |t_L^2| + |t_R^2|$, and T_K becomes 1.7 K. This also supports that the system is in the Kondo regime.

Comment 9. Eq. 1: the authors write ‘ G_S is the maximum value when $t_L = t_R$ and is equal to $2e^2/h$ in the case of $S = 1/2$ Kondo state’. This is confusing. Could the authors just say: ‘where G_S is the quantum of conductance $2e^2/h$. G_0 is maximum when $t_L = t_R$ ’

Reply: Thank you for this comment. We have corrected the sentence.

Changes(p. 8, l. 134)

“ t_L and t_R are the transmission coefficients from the dot to the left and right reservoir, respectively. G_S is the maximum value when $t_L = t_R$ and is equal to $2e^2/h$ in the case of $S = 1/2$ Kondo state”
is now

“ t_L and t_R are the transmission coefficients from the dot to the left and right reservoir, respectively, and G_S is the quantum of conductance $2e^2/h$. G_0 is maximum when $t_L = t_R$ ”

Comment 10. The authors extract T_K as a parameter from eqs. 2 and 3. Could they extract T_K from the FWHM of the zero-bias peak at base temperature and compare the values obtained with eqs. 2 and 3?

Reply: We appreciate the referee's comment. We agree and conducted the analysis from the FWHM. The FWHM of the zero-bias peak at $T = 56$ mK in Fig. 3(f) is 0.15 meV, and T_K is given by

$$T_K = e \cdot \text{FWHM} / k_B. \quad (4)$$

By using eq.(4), T_K is calculated to be 1.70 K. This result is consistent with the calculation of Comment 8. Kondo temperature given by s-free fitting in Fig. 3h is 2.27 K, and this is approximately consistent with T_K calculated by FWHM.

Following the referee's comment, we have added the following sentences in the text.

Added(p. 8, l. 139)

Moreover, we calculated T_K by using full width at half maximum (FWHM) of the zero-bias peak at $T = 56$ mK in Fig.3f. In earlier studies, the relation $T_K = e \cdot \text{FWHM}/k_B$ was used to estimate T_K [16]. By using this relation T_K was calculated to be 1.70 K, which is consistent with the calculation by Eq.1 discussed above.

Comment 11. *Page 11, paragraph 2: The authors eliminate small orbital energies compared to K_{BT} as an explanation for the breakdown of the even-odd behavior of the Kondo effect. This is well explained in Von Klitzing's paper as pointed out by the authors. As I explained in point 7, the analysis to extract orbital energy seems weak, so unless they can prove me wrong, they can't rule out the explanation by which the addition of electrons follow Hund's rule and therefore parallel spin configurations are favorable due to exchange interaction, deviating from the usual even-odd behavior.*

Reply: We thank the referee for this comment. As we have discussed in the reply to the referee's point 7, the orbital energy is estimated in two distinct ways and found consistent. Nevertheless, as noted by the referee, we still cannot stringently disprove various possibilities involving triplet state owing to the large correlation energy of ZnO. Therefore, we agree with the referee that we leave these possibilities for future studies. We have modified the expressions to reflect these possibilities.

Changes(p. 11, l. 187)

“Having considered these possibilities, we propose that the observed even-odd independent Kondo effect may involve multiple orbitals strongly hybridized with each other as indicated by the complex peak structures in Figs. 4.”

is now

“Having considered these possibilities, we also propose another mechanism of the observed even-odd independent Kondo effect that involves multiple orbitals strongly hybridized with each other as indicated by the complex peak structures in Figs. 4.”

Added(p. 11, l.199)

Nevertheless, we cannot completely rule out the possibility of the singlet-triplet or $S = 1$ Kondo effect because of the limitation of the detailed state analysis.

Response to Referee #2

We are grateful to the referee for careful and critical reading of our manuscript. We have revised our manuscript according to the comments. Our responses to the comments are described below.

Summary comment. *The paper by Noro et al. introduces an electrostatic approach to forming quantum dots in ZnO heterostructures, as previously has been demonstrated in, e.g., GaAs and Si systems. In electron transport measurements, manifestations of the Kondo effect are observed, however without clear even-odd electron number parity. The authors suggest that their work not only sheds light on correlated electron physics in quantum dots, but they also propose potential applications in quantum devices, capitalizing on ZnO's unique properties.*

The work represents a solid and very interesting study of the Kondo effect in quantum dots. The claimed impact for quantum devices in general, and spin qubits in particular could be better explained and substantiated in the paper.

Reply: We thank the reviewer for summarizing the critical points of our work.

Comment 1. *I wonder why only two full Coulomb diamonds are studied in Fig. 2e, which in my opinion makes it difficult to make strong and general statements about odd-even asymmetry in this system. Do the authors have data showing more consecutive Coulomb diamonds, which could for example be included in the Supplementary Material? In how many QD devices similar effects have been observed?*

Reply: We thank the referee for this comment. When changing the plunger gate voltage, the tunnel rate of quantum dots in ZnO changes significantly as shown in Fig. S1(a), which may reflect the character of ZnO. For this reason, the coupling between the QD and the source-drain electrodes changes quickly from strong to weak. This characteristic makes it difficult to observe Coulomb diamonds in a wide range of plunger gate voltages. Therefore, we confirmed the absence of even-odd parity in the Kondo effect by observing in two other QD devices. To explain this, we have added the following sentences in the text.

Added(p.7, l.114)

Because the tunnel rate changes significantly with the change in plunger gate voltage, the coupling between the QD and the source-drain electrodes changes quickly from strong to weak, which makes it hard to observe Coulomb diamonds in a wide range of gate voltages. We confirmed the absence of even-odd parity in the Kondo effect as observed in other devices(Supply, Fig.S1).

Comment 2. In Fig. 3h, could the authors compare their results with the exact NRG results for $S=1/2$ and $S=1$ in a G/G_0 vs T/T_K plot? The authors should also compare their results with the predictions for a two-stage Kondo effect for $S \geq 1$, as described in *Phys. Rev. Lett.* 88, 126803 (2002).

Reply: Thank you for this comment. We compared the results with the NRG results for $S = 1/2$ and $S = 1$ in a G/G_0 vs T/T_K plot [N. Roch *et al*, *Phys. Rev. Lett.* **103**, 197202 (2009)]. The following graph shows the result. Our data and empirical fitting by making s a free parameter look different from both NRG results of $S = 1/2$ and $S = 1$. For further discussion, detailed measurements in various conditions and comparison with the theoretical result using NRG will be useful and we think these are future research perspectives.

Reference: N. Roch *et al*, *Phys. Rev. Lett.* **103**, 197202 (2009).

We also considered the two-stage Kondo effect [*Phys. Rev. Lett.* 88, 126803 (2002)]. In the previous study, the conductance decrease was observed in zero bias peak of Kondo effect [Fig. 3 in ref.]. However, this dip wasn't observed in our study. According to the previous study, the first and second stage Kondo effect is characterized by T_{K1} and T_{K2} ($T_{K2} < T_{K1}$), respectively. When $(T, Vsd) < T_{K2} < T_{K1}$, the second stage of the Kondo effect quenches the first one, which makes the dip in the zero bias peak. Because we have not observed the dip structure, it is difficult to compare the result. Also from these, we agree that we cannot completely rule out the possibility of the $S = 1$ Kondo effect. To clarify these points, we modified the manuscript as follows.

Changes(p. 11, l. 187)

“Having considered these possibilities, we propose that the observed even-odd independent Kondo effect may involve multiple orbitals strongly hybridized with each other as indicated by the complex peak structures in Figs. 4.”

is now

“Having considered these possibilities, we also propose another mechanism of the observed even-odd independent Kondo effect that involves multiple orbitals strongly hybridized with each other as indicated by the complex peak structures in Figs. 4.”

Added(p. 11, l.199)

“Nevertheless, we cannot completely rule out the possibility of the singlet-triplet or $S = 1$ Kondo effect because of the limitation of the detailed state analysis.”

Added(p.9, l. 168)

“Future further measurements in various conditions and comparison of the results with numerical renormalization group (NRG) calculations [55-57] will contribute to understanding the detailed temperature dependence.”

Added(References)

[56]Roch, N. *et al*, *Phys. Rev. Lett.* **103**, 197202 (2009).

[57]Takada, S. *et al*, *Phys. Rev. Lett.* **113**, 126601 (2014).

Comment 3. *The authors should define the electron interaction parameter r_s on p3.*

Reply: We appreciate the referee's comment. The interaction parameter r_s is defined as the ratio of the Coulomb energy to the Kinetic energy, and is defined as

$$r_s = m^* e^2 / 4\pi \hbar^2 \varepsilon \sqrt{n\pi},$$

where m^* is the effective mass, e is the elementary electric charge, \hbar is the Planck constant divided by 2π , ε is the dielectric constant, and n is the sheet carrier density. Accordingly, we have added the definition of the r_s as follows.

Changed(p. 2, l. 41)

“Figure 1a summarizes the feature of ZnO compared to other semiconductor materials in terms of the electron interaction parameter r_s and the transport scattering time τ .”

is now

" Figure 1a summarizes the feature of ZnO compared to other semiconductor materials in terms of the electron interaction parameter r_s and the transport scattering time τ , where r_s is defined as the

ratio of the Coulomb energy to the Kinetic energy, and is expressed as $r_s = m^* e^2 / 4\pi\hbar^2 \epsilon \sqrt{n\pi}$ (m^* : effective mass, e : elementary electric charge, \hbar : Planck constant divided by 2π , ϵ : dielectric constant, n : sheet carrier density).

Comment 4. *p.9 typo: singlet-triplet*

Reply: Thank you for providing this comment. We have corrected this point.

Changes(p. 9, l. 173)

“Instead, the even-electron Kondo effect is realized at the single-triplet level degeneracy under a magnetic field [17, 20].”

is now corrected to

“Instead, the even-electron Kondo effect is realized at the singlet-triplet level degeneracy under a magnetic field [17, 20].”

Comment 5. *p.10 typo in caption Fig. 4: without*

Reply: We thank the referee for this comment. We have corrected this point.

Changes(Fig. 4 e,f caption)

“Schematic diagrams of spin filling and Kondo state in the case of ZnO without and with a magnetic field. ”

is now corrected to

“Schematic diagrams of spin filling and Kondo state in the case of ZnO without and with a magnetic field.”

Response to Referee #1

We are grateful to the referee for careful and critical reading of our manuscript.

Comment. *I have carefully reviewed the response from the authors. They have addressed adequately all my comments. In particular, establishing the energies using the co-tunneling signal corroborates their previous interpretation.*

I therefore recommend publication in Nature Communications of this improved version of the manuscript.

Reply: We thank the referee for recommending our revised manuscript to be published in Nature Communications.

Response to Referee #2

We are grateful to the referee for careful and critical reading of our manuscript. Our responses to the comments are described below.

Comment 1. *The authors have addressed the points I raised about the original version of the manuscript.*

(1) Multiple Coulomb diamonds and observation in other devices The authors explain convincingly that the coupling between the QD and the source-drain electrodes changes quickly from strong to weak when sweeping the plunger gate voltage. I understand that it is therefore difficult to study several consecutive Coulomb diamonds in a single device.

The authors do supply data of two other devices in the Supplementary Information. The data are not very clear (especially the resolution of Fig. S1b is rather low), but there seem to be zero-bias peaks in neighbouring Coulomb diamonds. Based on the provided data, it actually looks like there is always a zero-bias peak once the conductance (coupling) is high enough.

Reply: We thank the referee for this comment. As the referee mentioned, even though the data are not high resolution, zero-bias peaks are seen in Coulomb diamonds in Fig. S1. We agree with the referee that this confirms our claim.

Comment 2. *(2) Comparison with NRG results/two-stage Kondo effect The authors compared some of their results with the NRG results for $S = 1/2$ and $S = 1$ in a G/G_0 vs T/TK plot. However, it is not clear to me what Coulomb diamond they used for this analysis, and whether they found the same results in other (neighbouring) diamonds. This is a critical omission.*

The presented data can clearly not be fitted by NRG results for $S = 1/2$ nor $S = 1$. I therefore do not understand the remark “Nevertheless, we cannot completely rule out the possibility of the singlet-triplet or $S = 1$ Kondo effect because of the limitation of the detailed state analysis.” (line 199-200).

Indeed, I would argue that based on the mismatch with the NRG results, a more in-depth analysis is required where one would systematically investigate the role of (1) multi-channel Kondo effect, (2) spin-orbit coupling, (3) anisotropic Kondo effect, (4) higher spin states, (5) mixed-valence regime, (6) decoherence effects, (7) non-Fermi-liquid behaviour The authors adequately address the two-stage Kondo effect question.

I agree with the authors that “Future further measurements in various conditions and comparison of the results with numerical renormalization group (NRG) calculations[55–57] will contribute to

understanding the detailed temperature dependence.” (line 168-170). Given the amount of open questions, I find the claim for “parity-independent Kondo effect of correlated electrons” slightly premature and not very well substantiated. Although the work is definitely interesting, overall, the manuscript appears to me more as “work in progress”.

The results will be definitely of interest to a certain readership. Whether the manuscript in its present form is mature enough to warrant publication in *Nature Communications* is a judgement I leave to the Editor.

Reply: We appreciate the referee's comment. The data we used for comparison with NRG is the zero-bias peak at the electron number of $N+1$. The fitting with the empirical form cannot be applied to the zero-bias peaks for electron numbers of $N-1$ and N . This is because, from the temperature dependence shown in Fig. 3c and 3e, the conductance does not saturate at G_0 in the low temperature region. This makes it difficult to determine the values of G_0 and T_K . Therefore, it is not possible to compare the fitting in the cases of $N-1$ and N with NRG. We revised the figure in the response letter to indicate the electron number.

Comparison with NRG results

Reference: N. Roch *et al*, Phys. Rev. Lett. **103**, 197202 (2009)

As noted by the referee, our fitting results in Fig. 3h are not completely consistent with the parameters derived from NRG theories for $S = 1/2$ or $S = 1$ although they are somewhat closer to the case of $S = 1$. We also understand that there are still open questions about the detailed mechanism

in our observation of the Kondo effect in ZnO. We believe that this is partly because of the unique material parameters of ZnO compared with those of other semiconductors used to study the Kondo effect in quantum dots. In particular, the large effective mass, small dielectric constant, and large electron interaction parameter make ZnO an exceptional material with strong correlations, leading to several remarkable phenomena observed so far. In the present study, we agree that it would be ideal to reveal the complete physical picture of the observations by evaluating various contributions such as those (1) to (7) suggested by the referee and more. However, to complete the study, we would require a significant number of systematic sample sets and extensive measurements, which could correspond to several years of research and several intriguing topics for papers. Given these practical limitations, we believe that our results are innovative enough to reveal the existence of interesting issues in the Kondo effect, which may have been enabled by employing ZnO as a host material with unexplored ranges of physical parameters.

We believe that interesting open questions are necessary for the progress of science, and the finding of such questions itself is worth bringing to the attention of readers. Our present findings on the first gate-defined ZnO quantum dots with the novel material and parameters represent these interesting questions, and this will be supported by the referee's comment, "*The results will be definitely of interest to a certain readership,*" and by another referee's recommendation for publication.

Regarding our claim of the "parity-independent Kondo effect of correlated electrons," while the direct relationship between the independence of parity and the correlation effect has not been quantitatively clarified, many aspects of the strong correlation have been demonstrated in the 2DEG of ZnO. The correlation effect is most likely valid in the present study as well, because a strong exchange interaction is necessary for a finite S irrespective of electron parity. Therefore, we think that the title "parity-independent Kondo effect of correlated electrons" best represents the physical picture of our observation.